# Certified Defenses for Data Poisoning Attacks

**Jacob Steinhardt**[*]
Stanford University
jsteinha@stanford.edu

**Pang Wei Koh**[*]
Stanford University
pangwei@cs.stanford.edu

**Percy Liang**
Stanford University
pliang@cs.stanford.edu

## Abstract

Machine learning systems trained on user-provided data are susceptible to *data poisoning* attacks, whereby malicious users inject false training data with the aim of corrupting the learned model. While recent work has proposed a number of attacks and defenses, little is understood about the worst-case loss of a defense in the face of a determined attacker. We address this by constructing approximate upper bounds on the loss across a broad family of attacks, for defenders that first perform outlier removal followed by empirical risk minimization. Our approximation relies on two assumptions: (1) that the dataset is large enough for statistical concentration between train and test error to hold, and (2) that outliers within the clean (non-poisoned) data do not have a strong effect on the model. Our bound comes paired with a candidate attack that often nearly matches the upper bound, giving us a powerful tool for quickly assessing defenses on a given dataset. Empirically, we find that even under a simple defense, the MNIST-1-7 and Dogfish datasets are resilient to attack, while in contrast the IMDB sentiment dataset can be driven from 12% to 23% test error by adding only 3% poisoned data.

## 1 Introduction

Traditionally, computer security seeks to ensure a system's integrity against attackers by creating clear boundaries between the system and the outside world (Bishop, 2002). In machine learning, however, the most critical ingredient of all–the training data–comes directly from the outside world. For a system trained on user data, an attacker can inject malicious data simply by creating a user account. Such *data poisoning* attacks require us to re-think what it means for a system to be secure.

The focus of the present work is on data poisoning attacks against classification algorithms, first studied by Biggio et al. (2012) and later by a number of others (Xiao et al., 2012; 2015b; Newell et al., 2014; Mei and Zhu, 2015b; Burkard and Lagesse, 2017; Koh and Liang, 2017). This body of work has demonstrated data poisoning attacks that can degrade classifier accuracy, sometimes dramatically. Moreover, while some defenses have been proposed against specific attacks (Laishram and Phoha, 2016), few have been stress-tested against a determined attacker.

Are there defenses that are robust to a large class of data poisoning attacks? At development time, one could take a clean dataset and test a defense against a number of poisoning strategies on that dataset. However, because of the near-limitless space of possible attacks, it is impossible to conclude from empirical success alone that a defense that works against a known set of attacks will not fail against a new attack.

In this paper, we address this difficulty by presenting a framework for studying the entire space of attacks against a given defense. Our framework applies to defenders that (i) remove outliers residing outside a feasible set, then (ii) minimize a margin-based loss on the remaining data. For such defenders, we can generate approximate upper bounds on the efficacy of any data poisoning attack, which hold modulo two assumptions—that the empirical train and test distribution are close together,

---

[*]Equal contribution.

and that the outlier removal does not significantly change the distribution of the clean (non-poisoned) data; these assumptions are detailed more formally in Section 3. We then establish a duality result for our upper bound, and use this to generate a candidate attack that nearly matches the bound. Both the upper bound and attack are generated via an efficient online learning algorithm.

We consider two different instantiations of our framework: first, where the outlier detector is trained independently and cannot be affected by the poisoned data, and second, where the data poisoning can attack the outlier detector as well. In both cases we analyze binary SVMs, although our framework applies in the multi-class case as well.

In the first setting, we apply our framework to an "oracle" defense that knows the true class centroids and removes points that are far away from the centroid of the corresponding class. While previous work showed successful attacks on the MNIST-1-7 (Biggio et al., 2012) and Dogfish (Koh and Liang, 2017) image datasets in the absence of any defenses, we show (Section 4) that no attack can substantially increase test error against this oracle—the $0/1$-error of an SVM on either dataset is at most $4\%$ against any of the attacks we consider, even after adding $30\%$ poisoned data.[1] Moreover, we provide certified upper bounds of $7\%$ and $10\%$ test error, respectively, on the two datasets. On the other hand, on the IMDB sentiment corpus (Maas et al., 2011) our attack increases classification test error from $12\%$ to $23\%$ with only $3\%$ poisoned data, showing that defensibility is very dataset-dependent: the high dimensionality and abundance of irrelevant features in the IMDB corpus give the attacker more room to construct attacks that evade outlier removal.

For the second setting, we consider a more realistic defender that uses the empirical (poisoned) centroids. For small amounts of poisoned data ($\leq 5\%$) we can still certify the resilience of MNIST-1-7 and Dogfish (Section 5). However, with more ($30\%$) poisoned data, the attacker can subvert the outlier removal to obtain stronger attacks, increasing test error on MNIST-1-7 to $40\%$—much higher than the upper bound of $7\%$ for the oracle defense. In other words, defenses that rely on the (potentially poisoned) data can be much weaker than their data-independent counterparts, underscoring the need for outlier removal mechanisms that are themselves robust to attack.

## 2   Problem Setting

Consider a prediction task from an input $x \in \mathcal{X}$ (e.g., $\mathbb{R}^d$) to an output $y \in \mathcal{Y}$; in our case we will take $\mathcal{Y} = \{-1, +1\}$ (binary classification) although most of our analysis holds for arbitrary $\mathcal{Y}$. Let $\ell$ be a non-negative convex loss function: e.g., for linear classification with the hinge loss, $\ell(\theta; x, y) = \max(0, 1 - y\langle\theta, x\rangle)$ for a model $\theta \in \Theta \subseteq \mathbb{R}^d$ and data point $(x, y)$. Given a true data-generating distribution $p^*$ over $\mathcal{X} \times \mathcal{Y}$, define the test loss as $\mathbf{L}(\theta) = \mathbf{E}_{(x,y)\sim p^*}[\ell(\theta; x, y)]$.

We consider the *causative attack* model (Barreno et al., 2010), which consists of a game between two players: the *defender* (who seeks to learn a model $\theta$), and the *attacker* (who wants the learner to learn a bad model). The game proceeds as follows:

- $n$ data points are drawn from $p^*$ to produce a clean training dataset $\mathcal{D}_c$.
- The attacker adaptively chooses a "poisoned" dataset $\mathcal{D}_p$ of $\epsilon n$ poisoned points, where $\epsilon \in [0, 1]$ parametrizes the attacker's resources.
- The defender trains on the full dataset $\mathcal{D}_c \cup \mathcal{D}_p$ to produce a model $\hat{\theta}$, and incurs test loss $\mathbf{L}(\hat{\theta})$.

The defender's goal is to minimize the quantity $\mathbf{L}(\hat{\theta})$ while the attacker's goal is to maximize it.

**Remarks.** We assume the attacker has full knowledge of the defender's algorithm and of the clean training data $\mathcal{D}_c$. While this may seem generous to the attacker, it is widely considered poor practice to rely on secrecy for security (Kerckhoffs, 1883; Biggio et al., 2014a); moreover, a determined attacker can often reverse-engineer necessary system details (Tramèr et al., 2016).

The causative attack model allows the attacker to add points but not modify existing ones. Indeed, systems constantly collect new data (e.g., product reviews, user feedback on social media, or insurance claims), whereas modification of existing data would require first compromising the system.

Attacks that attempt to increase the overall test loss $\mathbf{L}(\hat{\theta})$, known as *indiscriminate availability* attacks (Barreno et al., 2010), can be thought of as a denial-of-service attack. This is in contrast to targeted

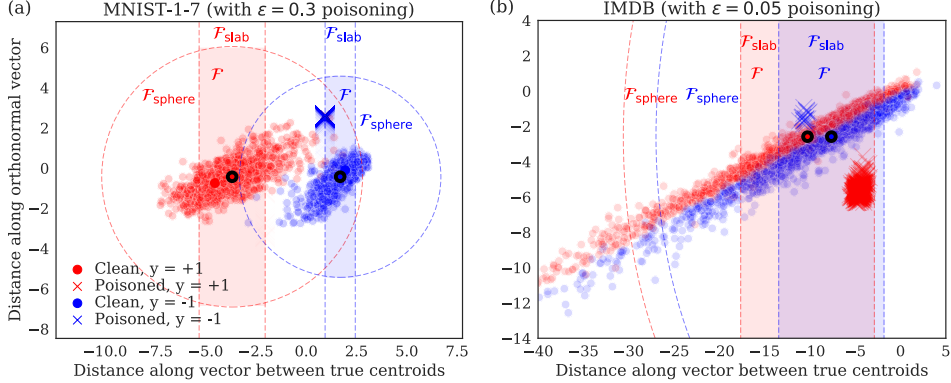

*Figure 1:* Different datasets possess very different levels of vulnerability to attack. Here, we visualize the effect of the sphere and slab oracle defenses, with thresholds chosen to match the 70th percentile of the clean data. We mark with an X our attacks for the respective values of $\epsilon$. **(a)** For the MNIST-1-7 dataset, the classes are well-separated and no attack can get past the defense. Note that our attack chooses to put all of its weight on the negative class here, although this need not be true in general. **(b)** For the IMDB dataset, the class centroids are not well-separated and it is easy to attack the classifier. See Section 4 for more details about the experiments.

attacks on individual examples or sub-populations (e.g., Burkard and Lagesse, 2017). Both have serious security implications, but we focus on denial-of-service attacks, as they compromise the model in a broad sense and interfere with fundamental statistical properties of learning algorithms.

## 2.1  Data Sanitization Defenses

A defender who trains naïvely on the full (clean + poisoned) data $\mathcal{D}_c \cup \mathcal{D}_p$ is doomed to failure, as even a single poisoned point can in some cases arbitrarily change the model (Liu and Zhu, 2016; Park et al., 2017). In this paper, we consider *data sanitization* defenses (Cretu et al., 2008), which examine the full dataset and try to remove the poisoned points, for example by deleting outliers. Formally, the defender constructs a *feasible set* $\mathcal{F} \subseteq \mathcal{X} \times \mathcal{Y}$ and trains only on points in $\mathcal{F}$:

$$\hat{\theta} \stackrel{\text{def}}{=} \underset{\theta \in \Theta}{\operatorname{argmin}} \, L(\theta; (\mathcal{D}_c \cup \mathcal{D}_p) \cap \mathcal{F}), \quad \text{where } L(\theta; S) \stackrel{\text{def}}{=} \sum_{(x,y) \in S} \ell(\theta; x, y). \qquad (1)$$

Given such a defense $\mathcal{F}$, we would like to upper bound the worst possible test loss over *any attacker* (choice of $\mathcal{D}_p$)—in symbols, $\max_{\mathcal{D}_p} \mathbf{L}(\hat{\theta})$. Such a bound would certify that the defender incurs at most some loss no matter what the attacker does. We consider two classes of defenses:

- *Fixed* defenses, where $\mathcal{F}$ does not depend on $\mathcal{D}_p$. One example for text classification is letting $\mathcal{F}$ be documents that contain only licensed words (Newell et al., 2014). Other examples are *oracle* defenders that depend on the true distribution $p^*$. While such defenders are not implementable in practice, they provide bounds: if even an oracle can be attacked, then we should be worried.
- *Data-dependent* defenses, where $\mathcal{F}$ depends on $\mathcal{D}_c \cup \mathcal{D}_p$. These defenders try to estimate $p^*$ from $\mathcal{D}_c \cup \mathcal{D}_p$ and thus are implementable in practice. However, they open up a new line of attack wherein the attacker chooses the poisoned data $\mathcal{D}_p$ to change the feasible set $\mathcal{F}$.

**Example defenses for binary classification.** Let $\mu_+ \stackrel{\text{def}}{=} \mathbb{E}[x \mid y = +1]$ and $\mu_- \stackrel{\text{def}}{=} \mathbb{E}[x \mid y = -1]$ be the centroids of the positive and negative classes. A natural defense strategy is to remove points that are too far away from the corresponding centroid. We consider two ways of doing this: the *sphere defense*, which removes points outside a spherical radius, and the *slab defense*, which first projects points onto the line between the centroids and then discards points that are too far on this line:

$$\mathcal{F}_{\text{sphere}} \stackrel{\text{def}}{=} \{(x, y) : \|x - \mu_y\|_2 \leq r_y\}, \quad \mathcal{F}_{\text{slab}} \stackrel{\text{def}}{=} \{(x, y) : |\langle x - \mu_y, \mu_y - \mu_{-y}\rangle| \leq s_y\}. \qquad (2)$$

Here $r_y, s_y$ are thresholds (e.g., chosen so that 30% of the data is removed). Note that both defenses are oracles ($\mu_y$ depends on $p^*$); in Section 5, we consider versions that estimate $\mu$ from $\mathcal{D}_c \cup \mathcal{D}_p$.

Figure 1 depicts both defenses on the MNIST-1-7 and IMDB datasets. Intuitively, the constraints on MNIST-1-7 make it difficult for an attacker, whereas IMDB looks far more attackable. In the next section, we will see how to make these intuitions concrete.

---

**Algorithm 1** Online learning algorithm for generating an upper bound and candidate attack.

---

**Input:** clean data $\mathcal{D}_c$ of size $n$, feasible set $\mathcal{F}$, radius $\rho$, poisoned fraction $\epsilon$, step size $\eta$.
Initialize $z^{(0)} \leftarrow 0$, $\lambda^{(0)} \leftarrow \frac{1}{\eta}$, $\theta^{(0)} \leftarrow 0$, $U^* \leftarrow \infty$.
**for** $t = 1, \ldots, \epsilon n$ **do**
    Compute $(x^{(t)}, y^{(t)}) = \operatorname{argmax}_{(x,y) \in \mathcal{F}} \ell(\theta^{(t-1)}; x, y)$.
    $U^* \leftarrow \min \left( U^*, \frac{1}{n} L(\theta^{(t-1)}; \mathcal{D}_c) + \epsilon \ell(\theta^{(t-1)}; x^{(t)}, y^{(t)}) \right)$.
    $g^{(t)} \leftarrow \frac{1}{n} \nabla L(\theta^{(t-1)}; \mathcal{D}_c) + \epsilon \nabla \ell(\theta^{(t-1)}; x^{(t)}, y^{(t)})$.
    Update: $z^{(t)} \leftarrow z^{(t-1)} - g^{(t)}$, $\quad \lambda^{(t)} \leftarrow \max(\lambda^{(t-1)}, \frac{\|z^{(t)}\|_2}{\rho})$, $\quad \theta^{(t)} \leftarrow \frac{z^{(t)}}{\lambda^{(t)}}$.
**end for**
**Output:** upper bound $U^*$ and candidate attack $\mathcal{D}_p = \{(x^{(t)}, y^{(t)})\}_{t=1}^{\epsilon n}$.

---

## 3 Attack, Defense, and Duality

Recall that we are interested in the worst-case test loss $\max_{\mathcal{D}_p} \mathbf{L}(\hat{\theta})$. To make progress, we consider three approximations. First, (i) we pass from the test loss to the training loss on the clean data, and (ii) we consider the training loss on the full (clean + poisoned) data, which upper bounds the loss on the clean data due to non-negativity of the loss. For any model $\theta$, we then have:

$$\mathbf{L}(\theta) \stackrel{(i)}{\approx} \frac{1}{n} L(\theta; \mathcal{D}_c) \stackrel{(ii)}{\leq} \frac{1}{n} L(\theta; \mathcal{D}_c \cup \mathcal{D}_p). \tag{3}$$

The approximation (i) could potentially be invalid due to overfitting; however, if we regularize the model appropriately then we can show that train and test are close by standard concentration arguments (see Appendix B for details). Note that (ii) is always a valid upper bound, and will be relatively tight as long as the model ends up fitting the poisoned data well.

For our final approximation, we (iii) have the defender train on $\mathcal{D}_c \cup (\mathcal{D}_p \cap \mathcal{F})$ (i.e., it uses the entire clean data set $\mathcal{D}_c$ rather than just the inliers $\mathcal{D}_c \cap \mathcal{F}$). This should not have a large effect as long as the defense is not too aggressive (i.e., as long as $\mathcal{F}$ is not so small that it would remove important points from the clean data $\mathcal{D}_c$). We denote the resulting model as $\tilde{\theta}$ to distinguish it from $\hat{\theta}$.

Putting it all together, the worst-case test loss from any attack $\mathcal{D}_p$ with $\epsilon n$ elements is approximately upper bounded as follows:

$$\max_{\mathcal{D}_p} \mathbf{L}(\hat{\theta}) \stackrel{(i)}{\approx} \max_{\mathcal{D}_p} \frac{1}{n} L(\hat{\theta}; \mathcal{D}_c) \stackrel{(ii)}{\leq} \max_{\mathcal{D}_p} \frac{1}{n} L(\hat{\theta}; \mathcal{D}_c \cup (\mathcal{D}_p \cap \mathcal{F}))$$
$$\stackrel{(iii)}{\approx} \max_{\mathcal{D}_p} \frac{1}{n} L(\tilde{\theta}; \mathcal{D}_c \cup (\mathcal{D}_p \cap \mathcal{F}))$$
$$= \max_{\mathcal{D}_p \subseteq \mathcal{F}} \min_{\theta \in \Theta} \frac{1}{n} L(\theta; \mathcal{D}_c \cup \mathcal{D}_p) \stackrel{\text{def}}{=} \mathbf{M}. \tag{4}$$

Here the final step is because $\tilde{\theta}$ is chosen to minimize $L(\theta; \mathcal{D}_c \cup (\mathcal{D}_p \cap \mathcal{F}))$. The *minimax loss* $\mathbf{M}$ defined in (4) is the central quantity that we will focus on in the sequel; it has duality properties that will yield insight into the nature of the optimal attack. Intuitively, the attacker that achieves $\mathbf{M}$ is trying to maximize the loss on the full dataset by adding poisoned points from the feasible set $\mathcal{F}$.

The approximations (i) and (iii) define the assumptions we need for our certificates to hold; as long as both approximations are valid, $\mathbf{M}$ will give an approximate upper bound on the worst-case test loss.

### 3.1 Fixed Defenses: Computing the Minimax Loss via Online Learning

We now focus on computing the minimax loss $\mathbf{M}$ in (4) when $\mathcal{F}$ is not affected by $\mathcal{D}_p$ (fixed defenses). In the process of computing $\mathbf{M}$, we will also produce candidate attacks. Our algorithm is based on no-regret online learning, which models a game between a learner and nature and thus is a natural fit to our data poisoning setting. For simplicity of exposition we assume $\Theta$ is an $\ell_2$-ball of radius $\rho$.

Our algorithm, shown in Algorithm 1, is very simple: in each iteration, it alternates between finding the worst attack point $(x^{(t)}, y^{(t)})$ with respect to the current model $\theta^{(t-1)}$ and updating the model in the direction of the attack point, producing $\theta^{(t)}$. The attack $\mathcal{D}_p$ is the set of points thus found.

To derive the algorithm, we simply swap min and max in (4) to get an upper bound on $\mathbf{M}$, after which the optimal attack set $\mathcal{D}_{\mathrm{p}} \subseteq \mathcal{F}$ for a *fixed* $\theta$ is realized by a single point $(x, y) \in \mathcal{F}$:

$$\mathbf{M} \leq \min_{\theta \in \Theta} \max_{\mathcal{D}_{\mathrm{p}} \subseteq \mathcal{F}} \frac{1}{n} L(\theta; \mathcal{D}_{\mathrm{c}} \cup \mathcal{D}_{\mathrm{p}}) = \min_{\theta \in \Theta} U(\theta), \text{ where } U(\theta) \overset{\text{def}}{=} \frac{1}{n} L(\theta; \mathcal{D}_{\mathrm{c}}) + \epsilon \max_{(x,y) \in \mathcal{F}} \ell(\theta; x, y). \tag{5}$$

Note that $U(\theta)$ upper bounds $\mathbf{M}$ for any model $\theta$. Algorithm 1 follows the natural strategy of minimizing $U(\theta)$ to iteratively tighten this upper bound. In the process, the iterates $\{(x^{(t)}, y^{(t)})\}$ form a candidate attack $\mathcal{D}_{\mathrm{p}}$ whose induced loss $\frac{1}{n} L(\tilde{\theta}; \mathcal{D}_{\mathrm{c}} \cup \mathcal{D}_{\mathrm{p}})$ is a lower bound on $\mathbf{M}$. We can monitor the duality gap between lower and upper bounds on $\mathbf{M}$ to ascertain the quality of the bounds.

Moreover, since the loss $\ell$ is convex in $\theta$, $U(\theta)$ is convex in $\theta$ (regardless of the structure of $\mathcal{F}$, which could even be discrete). In this case, if we minimize $U(\theta)$ using any online learning algorithm with sublinear regret, the duality gap vanishes for large datasets. In particular (proof in Appendix A):

**Proposition 1.** *Assume the loss $\ell$ is convex. Suppose that an online learning algorithm (e.g., Algorithm 1) is used to minimize $U(\theta)$, and that the parameters $(x^{(t)}, y^{(t)})$ maximize the loss $\ell(\theta^{(t-1)}; x, y)$ for the iterates $\theta^{(t-1)}$ of the online learning algorithm. Let $U^* = \min_{t=1}^{\epsilon n} U(\theta^{(t)})$. Also suppose that the learning algorithm has regret $\mathrm{Regret}(T)$ after $T$ time steps. Then, for the attack $\mathcal{D}_p = \{(x^{(t)}, y^{(t)})\}_{t=1}^{\epsilon n}$, the corresponding parameter $\tilde{\theta}$ satisfies:*

$$\frac{1}{n} L(\tilde{\theta}; \mathcal{D}_c \cup \mathcal{D}_p) \leq \mathbf{M} \leq U^* \quad and \quad U^* - \frac{1}{n} L(\tilde{\theta}; \mathcal{D}_c \cup \mathcal{D}_p) \leq \frac{\mathrm{Regret}(\epsilon n)}{\epsilon n}. \tag{6}$$

Hence, any algorithm whose average regret $\frac{\mathrm{Regret}(\epsilon n)}{\epsilon n}$ is small will have a nearly optimal candidate attack $\mathcal{D}_{\mathrm{p}}$. There are many algorithms that have this property (Shalev-Shwartz, 2011); the particular algorithm depicted in Algorithm 1 is a variant of regularized dual averaging (Xiao, 2010). In summary, we have a simple learning algorithm that computes an upper bound on the minimax loss along with a candidate attack (which provides a lower bound). Of course, the minimax loss $\mathbf{M}$ is only an approximation to the true worst-case test loss (via (4)). We examine the tightness of this approximation empirically in Section 4.

## 3.2   Data-Dependent Defenses: Upper and Lower Bounds

We now turn our attention to data-dependent defenders, where the feasible set $\mathcal{F}$ depends on the data $\mathcal{D}_{\mathrm{c}} \cup \mathcal{D}_{\mathrm{p}}$ (and hence can be influenced by the attacker). For example, consider the slab defense (see (2)) that uses the empirical (poisoned) mean instead of the true mean:

$$\mathcal{F}_{\mathrm{slab}}(\mathcal{D}_{\mathrm{p}}) \overset{\text{def}}{=} \{(x, y) : |\langle x - \hat{\mu}_y(\mathcal{D}_{\mathrm{p}}), \hat{\mu}_y(\mathcal{D}_{\mathrm{p}}) - \hat{\mu}_{-y}(\mathcal{D}_{\mathrm{p}})\rangle| \leq s_y\}, \tag{7}$$

where $\hat{\mu}_y(\mathcal{D}_{\mathrm{p}})$ is the empirical mean over $\mathcal{D}_{\mathrm{c}} \cup \mathcal{D}_{\mathrm{p}}$; the notation $\mathcal{F}(\mathcal{D}_{\mathrm{p}})$ tracks the dependence of the feasible set on $\mathcal{D}_{\mathrm{p}}$. Similarly to Section 3.1, we analyze the minimax loss $\mathbf{M}$, which we can bound as in (5): $\mathbf{M} \leq \min_{\theta \in \Theta} \max_{\mathcal{D}_{\mathrm{p}} \subseteq \mathcal{F}(\mathcal{D}_{\mathrm{p}})} \frac{1}{n} L(\theta; \mathcal{D}_{\mathrm{c}} \cup \mathcal{D}_{\mathrm{p}})$.

However, unlike in (5), it is no longer the case that the optimal $\mathcal{D}_{\mathrm{p}}$ places all points at a single location, due to the dependence of $\mathcal{F}$ on $\mathcal{D}_{\mathrm{p}}$; we must jointly maximize over the full set $\mathcal{D}_{\mathrm{p}}$. To improve tractability, we take a continuous relaxation: we think of $\mathcal{D}_{\mathrm{p}}$ as a probability distribution with mass $\frac{1}{\epsilon n}$ on each point in $\mathcal{D}_{\mathrm{p}}$, and relax this to allow any probability distribution $\pi_{\mathrm{p}}$. The constraint then becomes $\mathrm{supp}(\pi_{\mathrm{p}}) \subseteq \mathcal{F}(\mathcal{D}_{\mathrm{p}})$ (where $\mathrm{supp}$ denotes the support), and the analogue to (5) is

$$\mathbf{M} \leq \min_{\theta \in \Theta} \tilde{U}(\theta), \text{ where } \tilde{U}(\theta) \overset{\text{def}}{=} \frac{1}{n} L(\theta; \mathcal{D}_{\mathrm{c}}) + \epsilon \max_{\mathrm{supp}(\pi_{\mathrm{p}}) \subseteq \mathcal{F}(\pi_{\mathrm{p}})} \mathbf{E}_{\pi_{\mathrm{p}}}[\ell(\theta; x, y)]. \tag{8}$$

This suggests again employing Algorithm 1 to minimize $\tilde{U}(\theta)$. Indeed, this is what we shall do, but there are a few caveats:

- The maximization problem in the definition of $\tilde{U}(\theta)$ is in general quite difficult. We will, however, solve a specific instance in Section 5 based on the sphere/slab defense described in Section 2.1.
- The constraint set for $\pi_{\mathrm{p}}$ is non-convex, so duality (Proposition 1) no longer holds. In particular, the average of two feasible $\pi_{\mathrm{p}}$ might not itself be feasible.

To partially address the second issue, we will run Algorithm 1, at each iteration obtaining a distribution $\pi_{\mathrm{p}}^{(t)}$ and upper bound $\tilde{U}(\theta^{(t)})$. Then, for *each* $\pi_{\mathrm{p}}^{(t)}$ we will generate a candidate attack by sampling $\epsilon n$ points from $\pi_{\mathrm{p}}^{(t)}$, and take the best resulting attack. In Section 4 we will see that despite a lack of rigorous theoretical guarantees, this often leads to good upper bounds and attacks in practice.

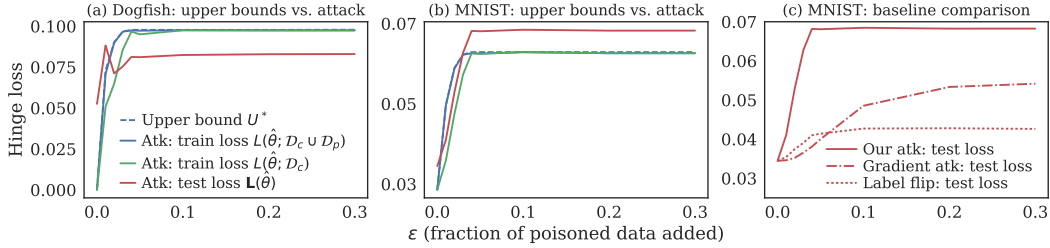

*Figure 2:* On the **(a)** Dogfish and **(b)** MNIST-1-7 datasets, our candidate attack (solid blue) achieves the upper bound (dashed blue) on the worst-case train loss, as guaranteed by Proposition 1. Moreover, this worst-case loss is low; even after adding 30% poisoned data, the loss stays below 0.1. **(c)** The gradient descent (dash-dotted) and label flip (dotted) baseline attacks are suboptimal under this defense, with test loss (red) as well as test error and train loss (not shown) all significantly worse than our candidate attack.

## 4 Experiments I: Oracle Defenses

An advantage of our framework is that we obtain a tool that can be easily run on new datasets and defenses to learn about the robustness of the defense and gain insight into potential attacks. We first study two image datasets: MNIST-1-7, and the Dogfish dataset used by Koh and Liang (2017). For MNIST-1-7, following Biggio et al. (2012), we considered binary classification between the digits 1 and 7; this left us with $n = 13007$ training examples of dimension 784. For Dogfish, which is a binary classification task, we used the same Inception-v3 features as in Koh and Liang (2017), so that each of the $n = 1800$ training images is represented by a 2048-dimensional vector. For this and subsequent experiments, our loss $\ell$ is the hinge loss (i.e., we train an SVM).

We consider the combined oracle slab and sphere defense from Section 2.1: $\mathcal{F} = \mathcal{F}_{\text{slab}} \cap \mathcal{F}_{\text{sphere}}$. To run Algorithm 1, we need to maximize the loss over $(x, y) \in \mathcal{F}$. Note that maximizing the hinge loss $\ell(\theta; x, y)$ is equivalent to minimizing $y\langle\theta, x\rangle$. Therefore, we can solve the following quadratic program (QP) for each $y \in \{+1, -1\}$ and take the one with higher loss:

$$\text{minimize}_{x \in \mathbf{R}^d} \; y\langle\theta, x\rangle \quad \text{subject to } \|x - \mu_y\|_2^2 \leq r_y^2, \quad |\langle x - \mu_y, \mu_y - \mu_{-y}\rangle| \leq s_y. \quad (9)$$

The results of Algorithm 1 are given in Figures 2a and 2b; here and elsewhere, we used a combination of CVXPY (Diamond and Boyd, 2016), YALMIP (Löfberg, 2004), SeDuMi (Sturm, 1999), and Gurobi (Gurobi Optimization, Inc., 2016) to solve the optimization. We plot the upper bound $U^*$ computed by Algorithm 1, as well as the train and test loss induced by the corresponding attack $\mathcal{D}_p$. Except for small $\epsilon$, the model $\tilde{\theta}$ fits the poisoned data almost perfectly. We think this is because all feasible attack points that can get past the defense can be easily fit without sacrificing the quality of the rest of the model; in particular, the model chooses to fit the attack points as soon as $\epsilon$ is large enough that there is incentive to do so.

The upshot is that, in this case, the loss $L(\tilde{\theta}; \mathcal{D}_c)$ on the clean data nearly matches its upper bound $L(\tilde{\theta}; \mathcal{D}_c \cup \mathcal{D}_p)$ (which in turn matches $U^*$). On both datasets, the certified upper bound $U^*$ is small ($< 0.1$ with $\epsilon = 0.3$), showing that the datasets are resilient to attack under the oracle defense.

We also ran the candidate attack from Algorithm 1 as well as two baselines — gradient descent on the test loss (varying the location of points in $\mathcal{D}_p$, as in Biggio et al. (2012) and Mei and Zhu (2015b)), and a simple baseline that inserts copies of points from $\mathcal{D}_c$ with the opposite label (subject to the flipped points lying in $\mathcal{F}$). The results are in in Figure 2c. Our attack consistently performs strongest; label flipping seems to be too weak, while the gradient algorithm seems to get stuck in local minima.[2] Though it is not shown in the figure, we note that the maximum test 0-1 error against any attack, for $\epsilon$ up to 0.3, was 4%, confirming the robustness suggested by our certificates.

Finally, we visualize our attack in Figure 1a. Interestingly, though the attack was free to place points anywhere, most of the attack is tightly concentrated around a single point at the boundary of $\mathcal{F}$.

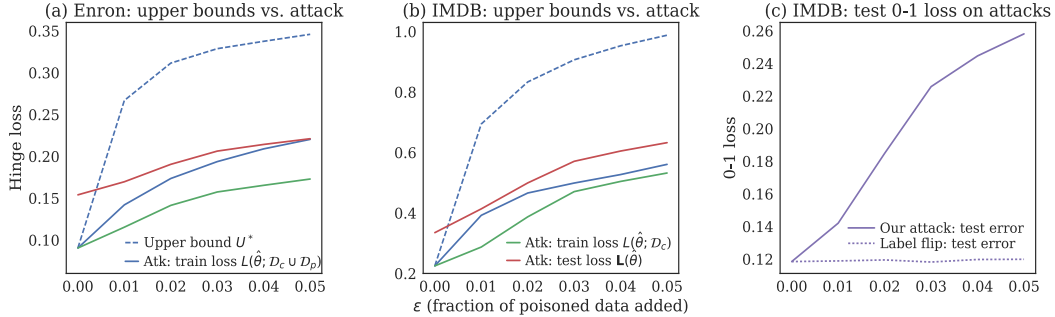

*Figure 3:* The **(a)** Enron and **(b)** IMDB text datasets are significantly easier to attack under the oracle sphere and slab defense than the image datasets from Figure 2. **(c)** In particular, our attack achieves a large increase in test loss (solid red) and test error (solid purple) with small $\epsilon$ for IMDB. The label flip baseline was unsuccessful as before, and the gradient baseline does not apply to discrete data. In (a) and (b), note the large gap between upper and lower bounds, resulting from the upper bound relaxation and the IQP/randomized rounding approximations.

## 4.1 Text Data: Handling Integrity Constraints

We next consider attacks on text data. Beyond the the sphere and slab constraints, a valid attack on text data must satisfy additional *integrity constraints* (Newell et al., 2014): for text, the input $x$ consists of binary indicator features (e.g., presence of the word "banana") rather than arbitrary reals.[3]

Algorithm 1 still applies in this case — the only difference is that the QP from Section 4 has the added constraint $x \in \mathbf{Z}_{\geq 0}^d$ and hence becomes an integer quadratic program (IQP), which can be computationally expensive to solve. We can still obtain upper bounds simply by relaxing the integrity constraints; the only issue is that the points $x^{(t)}$ in the corresponding attack will have continuous values, and hence don't correspond to actual text inputs. To address this, we use the IQP solver from Gurobi (Gurobi Optimization, Inc., 2016) to find an approximately optimal feasible $x$. This yields a valid candidate attack, but it might not be optimal if the solver doesn't find near-optimal solutions.

We ran both the upper bound relaxation and the IQP solver on two text datasets, the Enron spam corpus (Metsis et al., 2006) and the IMDB sentiment corpus (Maas et al., 2011). The Enron training set consists of $n = 4137$ e-mails (30% spam and 70% non-spam), with $d = 5166$ distinct words. The IMDB training set consists of $n = 25000$ product reviews with $d = 89527$ distinct words. We used bag-of-words features, which yields test accuracy 97% and 88%, respectively, in the absence of poisoned data. IMDB was too large for Gurobi to even approximately solve the IQP, so we resorted to a randomized rounding heuristic to convert the continuous relaxation to an integer solution.

Results are given in Figure 3; there is a relatively large gap between the upper bound and the attack. Despite this, the attacks are relatively successful. Most striking is the attack on IMDB, which increases test error from 12% to 23% for $\epsilon = 0.03$, despite having to pass the oracle defender.

To understand why the attacks are so much more successful in this case, we can consult Figure 1b. In contrast to MNIST-1-7, for IMDB the defenses place few constraints on the attacker. This seems to be a consequence of the high dimensionality of IMDB and the large number of irrelevant features, which increase the size of $\mathcal{F}$ without a corresponding increase in separation between the classes.

## 5 Experiments II: Data-Dependent Defenses

We now revisit the MNIST-1-7 and Dogfish datasets. Before, we saw that they were unattackable provided we had an oracle defender that knew the true class means. If we instead consider a data-dependent defender that uses the empirical (poisoned) means, how much can this change the attackability of these datasets? In this section, we will see that the answer is quite a lot.

As described in Section 3.2, we can still use our framework to obtain upper and lower bounds even in this data-dependent case, although the bounds won't necessarily match. The main difficulty is in computing $\tilde{U}(\theta)$, which involves a potentially intractable maximization (see (8)). However, for 2-class SVMs there is a tractable semidefinite programming algorithm; the full details are in

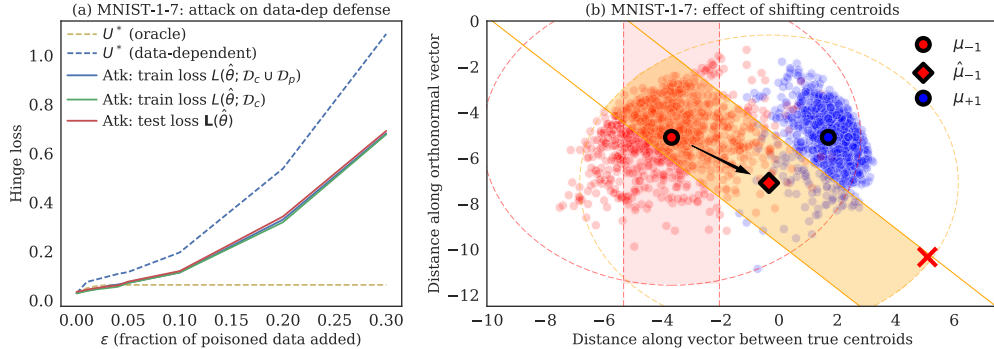

*Figure 4:* The data-dependent sphere and slab defense is significantly weaker than its oracle counterpart, allowing MNIST-1-7 and Dogfish to be successfully attacked. **(a)** On MNIST-1-7, our attack achieves a test loss of 0.69 (red) and error of 0.40 (not shown) at $\epsilon = 0.3$, more than $10\times$ its oracle counterpart (gold). At low $\epsilon \leq 0.05$, the dataset is safe, with a max train loss of 0.12. We saw qualitatively similar results on Dogfish. **(b)** Data-dependent sanitization can be significantly poisoned by coordinated adversarial data. We show here our attack for $\epsilon = 0.3$, which places almost all of its attacking mass on the red X. This shifts the empirical centroid, rotating the slab constraint (from red to orange) and allowing the red X to be placed far on the other side of the blue centroid.

Appendix D, but the rough idea is the following: we can show that the optimal distribution $\pi_p$ in (8) is supported on at most 4 points (one support vector and one non-support vector in each class). Moreover, for a fixed $\pi_p$, the constraints and objective depend only on inner products between a small number of points: the 4 attack points, the class means $\mu$ (on the clean data), and the model $\theta$. Thus, we can solve for the optimal attack locations with a semidefinite program on a $7 \times 7$ matrix. Then in an outer loop, we randomly sample $\pi_p$ from the probability simplex and take the one with the highest loss. Running this algorithm on MNIST-1-7 yields the results in Figure 4a. On the test set, our $\epsilon = 0.3$ attack leads to a hinge loss of 0.69 (up from 0.03) and a 0-1 loss of 0.40 (up from 0.01). Similarly, on Dogfish, our $\epsilon = 0.3$ attack gives a hinge loss of 0.59 (up from 0.05) and a 0-1 loss of 0.22 (up from 0.01).

The geometry of the attack is depicted in Figure 4b. By carefully choosing the location of the attack, the attacker can place points that lie substantially outside the original (clean) feasible set. This is because the poisoned data can substantially change the the direction of the slab constraint, while the sphere constraint by itself is not enough to effectively filter out attacks. There thus appears to be significant danger in employing data-dependent defenders—beyond the greater difficulty of analyzing them, they seem to actually be more vulnerable to attack.

# 6   Related Work

Due to their increased use in security-critical settings such as malware detection, there has been an explosion of work on the security of machine learning systems; see Barreno et al. (2010), Biggio et al. (2014a), Papernot et al. (2016b), and Gardiner and Nagaraja (2016) for some recent surveys.

Our contribution relates to the long line of work on data poisoning attacks; beyond linear classifiers, others have studied the LASSO (Xiao et al., 2015a), clustering (Biggio et al., 2013; 2014c), PCA (Rubinstein et al., 2009), topic modeling (Mei and Zhu, 2015a), collaborative filtering (Li et al., 2016), neural networks (Yang et al., 2017), and other models (Mozaffari-Kermani et al., 2015; Vuurens et al., 2011; Wang, 2016). There have also been a number of demonstrated vulnerabilities in deployed systems (Newsome et al., 2006; Laskov and Šrndič, 2014; Biggio et al., 2014b). We provide formal scaffolding to this line of work by supplying a tool that can certify defenses against a range of attacks.

A striking recent security vulnerability discovered in machine learning systems is *adversarial test images* that can fool image classifiers despite being imperceptible from normal images (Szegedy et al., 2014; Goodfellow et al., 2015; Carlini et al., 2016; Kurakin et al., 2016; Papernot et al., 2016a). These images exhibit vulnerabilities at test time, whereas data poisoning is a vulnerability at training time. However, recent adversarial attacks on reinforcement learners (Huang et al., 2017; Behzadan and Munir, 2017; Lin et al., 2017) do blend train and test vulnerabilities. A common defense against adversarial test examples is *adversarial training* (Goodfellow et al., 2015), which alters the training objective to encourage robustness.

We note that generative adversarial networks (Goodfellow et al., 2014), despite their name, are not focused on security but rather provide a game-theoretic objective for training generative models.

Finally, a number of authors have studied the theoretical question of learning in the presence of adversarial errors, under a priori distributional assumptions on the data. Robust algorithms have been exhibited for mean and covariance estimation and clustering (Diakonikolas et al., 2016; Lai et al., 2016; Charikar et al., 2017), classification (Klivans et al., 2009; Awasthi et al., 2014), regression (Nasrabadi et al., 2011; Nguyen and Tran, 2013; Chen et al., 2013; Bhatia et al., 2015) and crowdsourced data aggregation (Steinhardt et al., 2016). However, these bounds only hold for specific (sometimes quite sophisticated) algorithms and are focused on good asymptotic performance, rather than on giving good numerical error guarantees for concrete datasets/defenses.

## 7 Discussion

In this paper we have presented a tool for studying data poisoning defenses that goes beyond empirical validation by providing certificates against a large family of attacks modulo the approximations from Section 3. We stress that our bounds are meant to be used as a way to assess defense strategies in the design stage, rather than guaranteeing performance of a deployed learning algorithm (since our method needs to be run on the clean data, which we presumably would not have access to at deployment time). For instance, if we want to build robust defenses for image classifiers, we can assess the performance against attacks on a number of known image datasets, in order to gain more confidence in the robustness of the system that we actually deploy.

Having applied our framework to binary SVMs, there are a number of extensions we can consider: e.g., to other loss functions or to multiclass classification. We can also consider defenses beyond the sphere and slab constraints considered here—for instance, sanitizing text data using a language model, or using the covariance structure of the clean data (Lakhina et al., 2004). The main requirement of our framework is the ability to efficiently maximize $\ell(\theta; x, y)$ over all feasible $x$ and $y$. For margin-based classifiers such as SVMs and logistic regression, this only requires maximizing a linear function over the feasible set, which is often possible (e.g., via dynamic programming) even for discrete sets.

Our framework currently does not handle non-convex losses: while our method might still be meaningful as a way of generating attacks, our upper bounds would no longer be valid. The issue is that an attacker could try to thwart the optimization process and cause the defender to end up in a bad local minimum. Finding ways to rule this out without relying on convexity would be quite interesting.

Separately, the bound $\mathbf{L}(\hat{\theta}) \lesssim \mathbf{M}$ was useful because $\mathbf{M}$ admits the natural minimax formulation (5), but the worst-case $\mathbf{L}(\hat{\theta})$ can be expressed directly as a bilevel optimization problem (Mei and Zhu, 2015b), which is intractable in general but admits a number of heuristics (Bard, 1999). Bilevel optimization has been considered in the related setting of Stackelberg games (Brückner and Scheffer, 2011; Brückner et al., 2012; Zhou and Kantarcioglu, 2016), and is natural to apply here as well.

To conclude, we quote Biggio et al., who call for the following methodology for evaluating defenses:

> To pursue security in the context of an arms race it is not sufficient to *react* to observed attacks, but it is also necessary to *proactively anticipate* the adversary by *predicting* the most relevant, potential attacks through a what-if analysis; this allows one to develop suitable countermeasures *before* the attack actually occurs, according to the principle of *security by design*.

The existing paradigm for such proactive anticipation is to design various hypothetical attacks against which to test the defenses. However, such an evaluation is fundamentally limited because it leaves open the possibility that there is a more clever attack that we failed to think of. Our approach provides a first step towards surpassing this limitation, by not just anticipating but certifying the reliability of a defender, thus implicitly considering an infinite number of attacks before they occur.

**Reproducibility.** The code and data for replicating our experiments is available on GitHub (`http://bit.ly/gt-datapois`) and Codalab Worksheets (`http://bit.ly/cl-datapois`).

**Acknowledgments.** JS was supported by a Fannie & John Hertz Foundation Fellowship and an NSF Graduate Research Fellowship. This work was also partially supported by a Future of Life Institute grant and a grant from the Open Philanthropy Project. We are grateful to Daniel Selsam, Zhenghao Chen, and Nike Sun, as well as to the anonymous reviewers, for a great deal of helpful feedback.

## Footnotes

[1] We note Koh and Liang's attack on Dogfish targets specific test images rather than overall test error.

[2]Though Mei and Zhu (2015b) state that their cost is convex, they communicated to us that this is incorrect.

[3]Note that in the previous section, we ignored such integrity constraints for simplicity.

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
