[Supplementary Material]

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

## A  Proof of Proposition 1

Proposition 1 follows by standard duality arguments which we reproduce here. First recall the definition of Regret: for a sequence of loss functions $f_t(\theta)$, $t = 1, \ldots, T$, and an algorithm with iterates $\theta^{(1)}, \ldots, \theta^{(T)}$, regret is defined as

$$\text{Regret}(T) \stackrel{\text{def}}{=} \sum_{t=1}^{T} f_t(\theta^{(t)}) - \min_{\theta \in \Theta} \sum_{t=1}^{T} f_t(\theta). \tag{10}$$

In our particular case we take $f_t(\theta) = \frac{1}{n} L(\theta; \mathcal{D}_c) + \epsilon \ell(\theta; x^{(t+1)}, y^{(t+1)})$. Hence

$$\begin{aligned}
f_t(\theta^{(t)}) &= \frac{1}{n} L(\theta^{(t)}; \mathcal{D}_c) + \epsilon \ell(\theta^{(t)}; x^{(t+1)}, y^{(t+1)}) \\
&= \frac{1}{n} L(\theta^{(t)}; \mathcal{D}_c) + \epsilon \max_{(x,y) \in \mathcal{F}} \ell(\theta^{(t)}; x, y) = U(\theta^{(t)}).
\end{aligned} \tag{11}$$

Substituting into (10) and averaging over $T$, we have

$$\frac{\text{Regret}(T)}{T} = \frac{1}{T} \sum_{t=1}^{T} U(\theta^{(t)}) - \min_{\theta \in \Theta} \left( \frac{1}{n} L(\theta; \mathcal{D}_c) + \frac{\epsilon}{T} \sum_{t=1}^{T} \ell(\theta; x^{(t)}, y^{(t)}) \right). \tag{12}$$

For $t = \epsilon n$ the right-hand term is equal to $\frac{1}{n} L(\theta; \mathcal{D}_c \cap \{(x^{(t)}, y^{(t)})\}_{t=1}^{\epsilon n})$. Letting $\mathcal{D}_p = \{(x^{(t)}, y^{(t)})\}_{t=1}^{\epsilon n}$ and upper-bounding the min over $\theta$ by the value at $\tilde{\theta}$, we obtain

$$\frac{1}{n} L(\tilde{\theta}; \mathcal{D}_c \cup \mathcal{D}_p) \geq \frac{1}{T} \sum_{t=1}^{T} U(\theta^{(t)}) - \frac{\text{Regret } T}{T}, \tag{13}$$

and in particular $\frac{1}{n} L(\tilde{\theta}; \mathcal{D}_c \cup \mathcal{D}_p) \geq U^* - \frac{\text{Regret}(T)}{T}$, as was to be shown.

## B  Defending Against Overfitting Attacks

In Section 3 we claimed that it is possible to defend against overfitting attacks with appropriate regularization. In this section we justify this claim. The key is the classical theory of *uniform convergence*, which allows us to say that, with probability $1 - \delta$, the following uniform bound holds:

$$\left| \frac{1}{N} \sum_{(x,y) \in \mathcal{D}_c} \ell(\theta; x, y) - \mathbf{E}_{x,y \sim p^*}[\ell(\theta; x, y)] \right| \leq E(N, \rho, \delta), \tag{14}$$

where $E$ is an error bound that is roughly $\rho \sqrt{\frac{\log(1/\delta)}{N}}$. More precisely, Kakade et al. (2009) show the following:

**Theorem 1** (Corollary 5 of Kakade et al. (2009)). *Let $\ell(\theta; x, y)$ be any margin-based loss: $\ell(\theta; x, y) = \phi(y\langle \theta, x \rangle)$, where $\phi$ is 1-Lipschitz. Then the bound (14) holds with probability $1 - \delta$, for $E(N, \rho, \delta) = \rho R \left( \sqrt{\frac{4}{n}} + \sqrt{\frac{\log(1/\delta)}{2n}} \right)$, where $R$ is such that $\|x\|_2 \leq R$ with probability 1.*

By setting $\rho$ appropriately relative to $R$ and $n$ we can therefore guarantee that the train and test losses in (14) are close together, and therefore rule out any overfitting attack (because any attack that makes the test loss high would also have to make the train loss high).

## C  Regret Bound for Adaptive RDA

Our optimization algorithm (Algorithm 1) is similar in spirit to Regularized Dual Averaging (Xiao, 2010), but the known regret bounds for RDA do not apply directly because the regularizer is chosen adaptively to ensure the norm constraint $\|\theta\|_2 \leq \rho$ holds. In fact, a somewhat different analysis is required in this case, closer in spirit to that given by Steinhardt et al. (2014) for sparse linear regression. While the details would take us beyond the scope of this paper, we state the regret bound here:

**Theorem 2.** *After $T$ steps of the update in Algorithm 1, the regret of Algorithm 1 can be bounded as*

$$\text{Regret}(T) \leq \frac{\rho^2}{2\eta} + \sum_{t=1}^{T} \frac{\|g^{(t)}\|_2^2}{2\lambda_t}. \tag{15}$$

We make two observations: first, since $\lambda_t \geq \frac{1}{\eta}$ necessarily, by setting $\eta$ to be on the order of $\frac{1}{\sqrt{T}}$ we can ensure average regret $\mathcal{O}(1/\sqrt{T})$. On the other hand, in many instances $\lambda_t$ will actually increase linearly with $t$ (in order to enforce the norm constraints $\|\theta\|_2 \leq \rho$) in which case the average regret decreases at the faster rate $\mathcal{O}(\frac{\log(T)}{T})$. In either case, the average regret goes to 0 as $T \to \infty$.

# D    Semidefinite Program for $\tilde{U}(\theta)$

Here we elaborate on the semidefinite program for $\tilde{U}(\theta)$ that was discussed in Section 5. Recall the definition of $\tilde{U}(\theta)$:

$$\tilde{U}(\theta) = \frac{1}{n}L(\mathcal{D}_c) + \epsilon \max_{\text{supp}(\pi_p) \subseteq \mathcal{F}(\pi_p)} \mathbf{E}_{\pi_p}[\ell(\theta; x, y)]. \tag{16}$$

Our goal is to solve the maximization over $\pi_p$ in the special case that $\mathcal{F}$ is defined by the data-dependent sphere and slab defenses (with empirical centroids) and $\ell(\theta; x, y) = \max(1 - y\langle\theta, x\rangle, 0)$ is the hinge loss. First, we argue that the optimal $\pi_p$ without loss of generality is supported on at most four points $(x_{a,+}, 1)$, $(x_{b,+}, 1)$, $(x_{a,-}, -1)$, and $(x_{b,-}, -1)$, where the $x_a$ points are support vectors and the $x_b$ points are non-support vectors.

Indeed, suppose that there are two distinct support vectors which both lie in the positive class. Then replacing them both with their midpoint does not affect either $\mathcal{F}(\pi_p)$ or $\mathbf{E}_{\pi_p}[\ell(\theta; x, y)]$; moreover, since $\mathcal{F}(\pi_p)$ is convex for fixed $\pi_p$ both points are still feasible. A similar argument applies to the non-support vectors and to the negative class, so that indeed we may assume there are at most the four distinct points above in $\text{supp}(\pi_p)$.

Now, let $\pi_{a,+}$, $\pi_{a,-}$, $\pi_{b,+}$, and $\pi_{b,-}$ be the weights of these points under $\pi_p$. Letting $\mu_+$ and $\mu_-$ be the empirical means of the positive and negative class over $\mathcal{D}_c$, and $p_+$ and $p_-$ the empirical probability of the two classes, we have the following expression for $\hat{\mu}_y$:

$$\hat{\mu}_y(\pi_p) = \frac{p_y\mu_y + \pi_{a,y}x_{a,y} + \pi_{b,y}x_{b,y}}{p_y + \pi_{a,y} + \pi_{b,y}}. \tag{17}$$

Moreover, the objective $\mathbf{E}_{\pi_p}[\ell(\theta; x, y)]$ may be written as

$$\mathbf{E}_{\pi_p}[\ell(\theta; x, y)] = \pi_{a,+}(1 - \langle\theta, x_{a,+}\rangle) + \pi_{a,-}(1 + \langle\theta, x_{a,-}\rangle), \tag{18}$$

using the assumption that the $x_a$ are support vectors and the $x_b$ are not.

Now, the sphere and slab constraints may be written as

$$|\langle x_{i,y} - \hat{\mu}_y, \hat{\mu}_y - \hat{\mu}_{-y}\rangle| \leq s_y, \tag{19}$$

$$\langle x_{i,y} - \hat{\mu}_y, x_{i,y} - \hat{\mu}_y\rangle \leq r_y^2 \tag{20}$$

for $i \in \{a, b\}$, $y \in \{+1, -1\}$. We also have the constraints

$$1 - y\langle\theta, x_{a,y}\rangle \geq 0 \tag{21}$$

$$1 - y\langle\theta, x_{b,y}\rangle \leq 0 \tag{22}$$

for $y \in \{+1, -1\}$ (encoding the constraints that the $x_a$ are support vectors and the $x_b$ are not).

A careful examination reveals that, for fixed $\pi_{\{a,b\},\{+,-\}}$, all terms in (18-22) can be written as linear inequality constraints in the inner products between the 7 vectors $x_{a,+}, x_{a,-}, x_{b,+}, x_{b,-}, \mu_+, \mu_-, \theta$. Therefore, by changing variables to the $7 \times 7$ Gram matrix $G$ among these vectors, we can express the maximization over $\pi_p$ in (16) as a semidefinite program over these variables, with equality constraints for the known inner products between $\mu_+$, $\mu_-$, and $\theta$.

Moreover, for any matrix $G \succeq 0$ satisfying these equality constraints, it is possible to recover vectors $x_{a,+}$, $x_{a,-}$, $x_{b,+}$, and $x_{b,-}$ (depending on $\mu_+$, $\mu_-$, and $\theta$) whose inner products match the Gram

matrix $G$. Precisely, if $G = \begin{bmatrix} G_{11} & G_{12} \\ G_{21} & G_{22} \end{bmatrix}$ is the Gram matrix (with block 1 being the 4 vectors $\{x_{a,+}, x_{a,-}, x_{b,+}, x_{b,-}\}$, and block 2 being the 3 known vectors $\{\mu_+, \mu_-, \theta\}$), then for any vectors $\{v_{a,+}, v_{a,-}, v_{b,+}, v_{b,-}\}$ orthogonal to the span of $\mu_+$, $\mu_-$, and $\theta$, we can take

$$\begin{bmatrix} x_{a,+} & x_{a,-} & x_{b,+} & x_{b,-} \end{bmatrix} = \begin{bmatrix} v_{a,+} & v_{a,-} & v_{b,+} & v_{b,-} \end{bmatrix} A + \begin{bmatrix} \mu_+ & \mu_- & \theta \end{bmatrix} B, \qquad (23)$$

where $A^\top A = G_{11} - G_{12} G_{22}^\dagger G_{21}$ and $B = G_{22}^\dagger G_{21}$, and $\dagger$ denotes pseudoinverse. This means that solving the SDP allows us to not only compute the optimal objective value, but also to actually recover vectors $x$ realizing it.

To finish, we must handle the fact that the weights $\pi_{\{a,b\},\{+,-\}}$ are not known. However, they comprise only a 3-dimensional parameter space, and hence we can approximate the maximum over all $\pi_{\{a,b\},\{+,-\}}$ through Monte Carlo simulation (i.e., randomly sample the weights a sufficiently large number of times and take the best).