[Reviews · NeurIPS 2017]

Reviewer 1



The paper studies an interesting topic and is overall nice to read. However, I have doubts about the correctness of the main theorem (Proposition 1), which makes two statements: one that bounds the max-min loss M from below and above, and a second that further bounds the upper bound. While the second claim is proven in the appendix; the first is not. The upper bound follows from the convexity of U and Equation 5. (It might be helpful for the reader to explicitly state this finding and argue why D_p reduces to a set of \epsilon n identical points.) However, I can't see why the lower bound holds. Computing M requires the solution of a saddle point problem. If one knew one part of this saddle point solution, one could easily bound M from above and/or below. However, Algorithm 1 only generates interim solutions that eventually converge to the saddle point. Concretely, in (6), D_p is not (necessarily) part of the saddle point solution; and \hat theta is neither the minimizer of L w.r.t. D_c \cup D_p nor part of the saddle point solution. So why is this necessarily a lower bound? It's hard to judge the quality of candidate set D_p. One could argue that it would be better to actually run the algorithm till convergence and take the final data pair (x, y) \epsilon n times. Alternatively, one could take the last \epsilon n data points before the algorithm converges. Why the first \epsilon n update steps give a better candidate set is unclear to me. Whether the resulting model after \epsilon n update steps is any close to an optimum is unclear to me, too. Overall, there is little theoretical justification for the generation of the candidate attack in Alg. 1; it is for sure not the solution of (4). Hence, it's hard to conclude something like "even under a simple defense, the XYZ datasets are resilient to attack". And since the attack is model-based - it is constructed with a certain model (in this case, a linear model), loss function, and regularization scheme in mind - it is also hard to conclude "the ABC dataset can be driven from 12% to 23% test error"; this might be true for the concretely chosen model, but a different model might be more robust to this kind of poising attack. To me the value of the approach is that it allows to study the sensitivity of a concrete classifier (e.g., SVM) w.r.t. a rather broad set of attacks. In case of such sensitivity (for a concrete dataset), one could think about choosing a different defense model and/or classifier. However, in the absence of such sensitivity one should NOT conclude the robustness of the model. In Section 3.2, the defenses are defined in terms of class means rather than centroids (as in (2)). Eq. (8) requires F(\pi), though, F is defined over a set of data points. I don't fully understand this maximization term and I would like the authors to provide more details how to generalize Alg. 1 to this setting. In the experimentation section, some details are missing to reproduce the results; for example, the algorithm has parameters \roh and \eta, the defense strategies have thresholds. It is neither clear to me how those parameters are chosen, nor which parameters were used in the experiments. After authors response: Regarding the correctness of correctness of Prop. 1: \hat {theta} is the minimizer of L w.r.t. D_c \cup D_p; but D_c \cup D_p is not simultaneously a maximizer of L. M states the value of L at a saddle point, but neither \hat{theta} nor D_c \cup D_p is necessarily part of a saddle point. However, it became clear to me that this is not required. We can replace max_D min_\theta L(D, \theta) by max_D L(D, \hat {\theta}_D) where \hat{theta}_D is the minimizer w.r.t. D. Of course, this quantity is always smaller than M := L(\hat{D}, \hat{\theta}_\hat{D}) for any maximizer \hat{D} and saddle point (\hat{D}, \hat{\theta}_\hat{D}). I got confused from the two meanings of D_p, which is used as a free variable in the maximization in (4) and as the result of Alg. 1.

Reviewer 2



Title: 1988-Certified defenses for data poisoning attacks Summary: The paper proposes a method of finding approximate upper bound and constructing candidate attacks for poisoning attacks on simple defenders that solves a convex loss minimization problems. Strengths: The paper proposes original and rigorous approach toward an important problem. Experiments were performed carefully. Weaknesses: Not many, but there are several approximations made in the derivation. Also the strong assumptions of convexity and norm-based defense were made. Qualitative evaluation Quality: The paper is technically sound for the most part, except for severals approximation steps that perhaps needs more precise descriptions. Experiments seem to support the claims well. Literature review is broad and relevant. Clarity: The paper is written and organized very well. Perhaps the sections on data-dependent defenses are a bit complex to follow. Originality: While the components such as online learning by regret minimization and minimax duality may be well-known, the paper uses them in the context of poisoning attack to find a (approximate) solution/certificate for norm-based defense, which is original as far as I know. Significance: The problem addressed in the paper has been gaining importance, and the paper presents a clean analysis and experiments on computing an approximate upperbound on the risk, assuming a convex loss and/or a norm-based defense. Detailed comments I didn't find too many things to complain. To nitpick, there are two concerns about the paper. 1. The authors introduce several approximations (i -iii) which leaves loose ends. It is understandable that approximations are necessary to derive clean results, but the possible vulnerability (e.g., due to the assumption of attacks being in the feasible set only in lines 107-110) needs to be expanded to reassure the readers that it is not a real concern. 2. Secondly, what relevance does the framework have with problems of non-convex losses and/or non-norm type defenses? Would the non-vanishing duality gap and the difficulty of maximization over non-norm type constraints make the algorithm irrelevant? Or would it still give some intuitions on the risk upperbound? p.3, binary classification: If the true mean is known through an oracle, can one use the covariance or any other statistics to design a better defense?

Reviewer 3



Given some assumptions on the learning procedure, this manuscript bounds the effectiveness of data poisoning attacks on the learner's loss and introduces an attack that nearly achieves that bound in two settings: a data-independent and data-dependent defense. The problem considered is important, and while much of the community's efforts have been devoted to test time attacks (e.g., adversarial examples), somewhat less attention has been given to causative attacks. The methodology followed is appropriate given the security settings: bounding the power of an attacker given a realistic threat model is a first step towards preventing the often inevitable arms race between attackers and defenders. The writing could be improved by stressing two takeaways that are somewhat hidden in Sections 4 and 5. The first one is the observation made at lines 214-217. It seems like an important take-away from the experimental results and would benefit from being emphasized in the introduction and discussion. Likewise for the fact that data-dependent defenses are much less effective at mitigating data poisoning attacks than data-independent defenses. Although it may appear evident in hindsight, it is a useful observation for future efforts to build upon. Some of the weaknesses of the manuscript stem from the restrictive---but acceptable---assumptions made throughout the analysis in order to make it tractable. The most important one is that the analysis considers the impact of data poisoning on the training loss in lieu of the test loss. This simplification is clearly acknowledged in the writing at line 102 and defended in Appendix B. Another related assumption is made at line 121: the parameter space is assumed to be an l2-ball of radius rho. The paper is well written. Here are some minor comments: - The appendices are well connected to the main body, this is very much appreciated. - Figure 2 and 3 are hard to read on paper when printed in black-and-white. - There is a typo on line 237. - Although the related work is comprehensive, Section 6 could benefit from comparing the perspective taken in the present manuscript to the contributions of prior efforts. - The use of the terminology "certificate" in some contexts (for instance at line 267) might be misinterpreted, due to its strong meaning in complexity theory.